# Effects of Socio-Environmental Factors on Malaria Infection in Pakistan: A Bayesian Spatial Analysis

**DOI:** 10.3390/ijerph16081365

**Published:** 2019-04-16

**Authors:** Muhammad Farooq Umer, Shumaila Zofeen, Abdul Majeed, Wenbiao Hu, Xin Qi, Guihua Zhuang

**Affiliations:** 1School of Public Health, Xi’an Jiaotong University Health Science Center, Xi’an 710061, China; rafooq@hotmail.com (M.F.U.); shumailazofeen@yahoo.com (S.Z.); 2Directorate of Malaria Control Program, Islamabad 44000, Pakistan; amjaffar@hotmail.com; 3School of Public Health and Social Work, Queensland University of Technology, Kelvin Grove, QLD 4059, Australia; w2.hu@qut.edu.au; 4Global Health Institute, Xi’an Jiaotong University Health Science Center, Xi’an 710061, China

**Keywords:** Socio-climate variables, malaria, Pakistan, Bayesian CAR model

## Abstract

The role of socio-environmental factors in shaping malaria dynamics is complex and inconsistent. Effects of socio-environmental factors on malaria in Pakistan at district level were examined. Annual malaria cases data were obtained from Directorate of Malaria Control Program, Pakistan. Meteorological data were supplied by Pakistan Meteorological Department. A major limitation was the use of yearly, rather than monthly/weekly malaria data in this study. Population data, socio-economic data and education score data were downloaded from internet. Bayesian conditional autoregressive model was used to find the statistical association of socio-environmental factors with malaria in Pakistan. From 136/146 districts in Pakistan, >750,000 confirmed malaria cases were included, over a three years’ period (2013–2015). Socioeconomic status ((posterior mean value −3.965, (2.5% quintile, −6.297%), (97.5% quintile, −1.754%)) and human population density (−7.41 × 10^−4^, −0.001406%, −1.05 × 10^−4^ %) were inversely related, while minimum temperature (0.1398, 0.05275%, 0.2145%) was directly proportional to malaria in Pakistan during the study period. Spatial random effect maps presented that moderate relative risk (RR, 0.75 to 1.24) and high RR (1.25 to 1.99) clusters were scattered throughout the country, outnumbering the ones’ with low RR (0.23 to 0.74). Socio-environmental variables influence annual malaria incidence in Pakistan and needs further evaluation.

## 1. Introduction

Malaria incidence exhibited considerable decline worldwide since year 2000 [1,2]. Despite this noteworthy progress, 2014 onwards, the drop rate seems to be arrested or even reversed in some of the World Health Organization (WHO) regions [3]. This change in disease trend stresses on capitalizing decades long efforts, to achieving global goals of malaria control and elimination, with special focus on the developing countries in the world.

Socio-environmental variables influence spatial and temporal distribution, intensity and duration of infectious diseases in general [4,5,6]; and occurrence, transmission, seasonality and periodicity of malaria in particular [7,8,9]. Malaria dynamics are better understood in the perspective of native socioeconomic and climatic conditions to discover the role of local factors influencing malaria variability in space and time [10].

Meteorological factors, e.g., rainfall, temperature and humidity, are established to having associations with malaria incidence from temporal and spatial perspectives [9,11,12,13,14]. These meteorological factors when co-act synergistically, increase the duration of larvae development, shorten the incubation period of parasites, prolong mosquito survival, provide a favorable swampy habitat to the vectors, and increase number of mosquitoes and their bites, thus positively related to malaria high-risk [15,16,17,18].

Malaria and poverty have long been linked together; socioeconomic status along with education are believed to be indirectly related to malaria, and improvement in both these factors has contributed in disease control in many developing countries in recent times [19,20]. Human population density, like other socio-environmental factors contribute to heterogeneity in malaria infection, is indirectly related to entomological inoculation rate (EIR) thereby reducing the risk of malaria in areas with high human population densities [21].

Malaria is moderately endemic in Pakistan; yet, its transmission is unstable with disease burden ranging from very high (some high-risk districts in Federally Administered Tribal Areas (FATA) and Balochistan fear epidemic outbreak) to almost naught (some districts in Punjab and Azad Kashmir) [22]. Erratic malaria transmission patterns due to various factors like climatic changes, natural disasters (frequent flooding and irregular rainfall pattern), resource constraints, weak health infrastructure, poor socio-economic conditions, domestic unrest in many of the districts in FATA and Balochistan and limited access to healthcare delivery services and frequent migrations of people within the country and in the Pak-Afghanistan bordering area are amongst the key reasons for unstable malaria incidence in Pakistan [19]. According to recent estimates, 29% of the total population in Pakistan lives in malaria high transmission districts, while 69% in the districts with low malaria transmission [23]. There is need to explore in detail the malaria epidemiology in Pakistan, as meticulous studies based on modern statistical methods (e.g., Geographical Information System (GIS) based techniques), explaining dynamics of malaria in the country are barely adequate [22,24].

Bayesian modeling is a widely used mathematical technique in epidemiological studies. The outcomes from Bayesian model may have a variety of assistance in providing baseline for planning, resource allocation, monitoring and evaluation, identification of high-risk areas, estimation of disease burden and grading the progress aimed at control and elimination of malaria [9,25,26]. This study is first to be done in Pakistan in which, using Bayesian statistical methods, we intended to find out the effects of socio-environmental (socio-demographic such as human development index, education ranking, and population density; and meteorological variables such as temperature, rainfall, and relative humidity) on annual malaria incidence in Pakistan at the district level.

## 2. Methods

### 2.1. Data Collection

Annual, aggregated, laboratory confirmed malaria cases data (2013 to 2015) were collected from The Directorate of Malaria Control Program Pakistan (DoMC) via letter No. {F.No 6-1/2017/PMU/NFM(tech/RM)}. This data represented public sector healthcare facilities in 136 of total 146 districts in Pakistan (DoMC does not operate in 10 districts, i.e., Islamabad and districts in Gilgit Baltistan). Malaria cases were diagnosed through microscopy or rapid diagnostic tests (RDT) recommended by World Health Organization (WHO) [3], which is uniformly implemented in public sector healthcare facilities and remained consistent throughout the study period. Population census data 2017 were downloaded from Pakistan Bureau of Statistics (www.pbscensus.gov.pk), containing district-wise population as well as population density data. District-wise population data of 2017 was retrospectively extrapolated, depreciating by 2.4% annually (average annual growth rate of the country between last census done in 1998 and latest census done in 2017 is 2.4%) to get the population of 2015, 2014 and 2013 respectively. Socio-demographic variables included in this study were Human Development Index (HDI), district education score (ES) and population density (PD). HDI in our study was used to describe the socio-economic status for respective districts in the country. HDI is a summary measure of human development in three basic dimensions, i.e., health (life expectancy at birth), education (years of education received by people aged 25 years and older) and standard of living (measured by Gross National Income, GNI per capita). Experts consider HDI as arguably a better development indicator than Gross Domestic Product (GDP) as it explains how rich the lives of people are, rather than how rich the economy is. District wise, HDI were downloaded from United Nations Development Program Pakistan website (www.pk.undp.org). ES is the district-wise education score in Pakistan (arithmetic average of, (i) enrolment, (ii) learning outcomes, (iii) retention and (iv) gender parity scores; giving equal weight to each of these indicators) and is downloaded from “Alif Ailaan” website (www.alifailaan.pk/district_rankings), which is a non-profit organization working in the field of education in Pakistan. Meteorological variables data (RF, T(min.), T (max.), T (mean), RH, and WS) of 20 different stations across the country for years 2013, 2014 and 2015 were collected from Pakistan Meteorological Department, Islamabad, via letter No. (CDP-7(4)/3/A/2018). The data from these 20 stations were transferred into district level data by using Kriging interpolation (separately for each year 2013, 2014 and 2015). The annual average (2013–2015) of malaria incidence as well as meteorological variables were separately calculated from individual years’ data (2013, 2014 and 2015), to be utilized for analysis in current study.

### 2.2. Data Analysis

The vector-borne diseases exhibit periodicity due to endogenous and exogenous factors, and literature shows that periodicity due to extrinsic factors (such as socio-climatic factors) is shorter than that due to the intrinsic ones (factors intrinsic to the parasites) [27]; thus, we applied Bayesian statistical analysis on the three years’ combined annual average data so that this period was neither too short that it may mimic the results from any single year malaria data analysis (2013, 2014 or 2015), nor was too long that the periodicity due to extrinsic factors (socio-climatic factors) may be masked by the periodicity due to intrinsic factors.

In this study, we defined malaria incidence per 100,000 (annual average 2013–2015) as the dependent variable while socio-environmental variables as independent variables. The descriptive statistics and Spearman’s correlation were implemented in SPSS 13.0 (SPSS Inc., Chicago, IL, USA, 2005). The Bayesian conditional autoregressive (CAR) model has previously been used to describe spatial heterogeneity in the respective study areas in a specific disease risk [9,28]; we used a similar CAR model to explore the statistical association of socio-economic factors in malaria transmission across different districts in Pakistan. The formulation of Bayesian CAR model is as follows [29]:

log(*µi*) = log(*ni*) + (*β*0 + *β*1*X*1i + … + *β*m*X*mi) + *Ui* + *Si*(1)
In this formula; *Ui*, is the unstructured random effects and *Si*, the structured random effects, which are spatially correlated, and they represent a Poisson distribution with *µi* (mean of the dependent variable, i.e., malaria incidence). *i*, *n*, *X* and *β* represents location, population, the fixed effect and socio-environmental variables, respectively. *β*0 + *β*1*X*1*i* + … + *β*m*X*mi is the regression equation.

The Besag, York and Mollie (BYM) model was used to estimate the spatial risk patterns related to malaria by including fixed covariates along with their random effects. To begin with, we applied exploratory data analyses in the Bayesian CAR model; no spatial elements were added initially and then structured (*Si*) and unstructured covariance (*Ui*) were added. After testing different combinations of meteorological variables in Bayesian CAR models, the independent variables in the final model were selected based on multicollinearity (*rs* ≤ |0.80|) and the strength of their statistical association with malaria incidence, respectively. Amongst the meteorological variables, our final model comprised of RF, T (min.) and RH (T (max.), T (mean) and WS were not included), while the rest of the independent variables included in the final model were HDI, ES and PD. The smallest Deviance Information Criteria (DIC) value indicated the better goodness of fit and thus was considered as our final model for which residuals were mapped. Parameter estimation of the models was done by Markov chain Monte Carlo (MCMC) simulation technique using single chain algorithm. For each of the models, a burn-in of 30,000 iterations was considered followed by 90,000 iterations run. Autocorrelations of selected parameters were used to check the convergence. The Bayesian CAR model was run in the WinBUGS package 1.4 (Medical Research Council, Biostatistics Unit, Cambridge, UK). An exemption approval letter was obtained from National Bioethical Committee Pakistan via letter number No.4-87/NBC-279-Exempt./17/1139. The research protocol was approved by Institutional Review Board of Xi’an Jiaotong University, China.

## 3. Results

A total of 757,528 laboratory confirmed malaria cases were recorded in the study period (2013–2015) with 281,217, 275,149 and 201,162 cases in 2013, 2014 and 2015, respectively. The number of malaria cases relatively declined over the study period, despite the average annual population increase by 2.4% at the district level in Pakistan. Table 1 shows the descriptive statistics of dependent (annual malaria incidence) and independent variables (socio-environmental) used in this study. All the variables show wide disparities amongst the districts, e.g., malaria incidence across the districts varied from 0.04 to 11,565.23 with a standard deviation of 1239.88, and similar trends were seen for each of the independent variables in the study. This table also illustrated the differences in socioeconomic status and climatic conditions at the district level, throughout the country.

Figure 1 shows administrative units in Pakistan and will be called as provinces in this study. Figure 2 describes incidence rate of malaria at the district level in combined three years (2013–2015). This figure is a detailed description on the malaria incidence shown in Table 1, as it clearly indicates, low, medium, high and very high malaria endemic districts during the study period, appreciating the characteristic unstable nature of malaria infection in Pakistan. The map showed disease high-risk clusters in Balochistan, Sindh and FATA while the rest of the provinces had moderate to low malaria incidence in their respective districts.

Table 2 demonstrated correlation between malaria incidence and socio-environmental variables during the course of study. All three socio-demographic variables (HDI, ES and PD) were found negatively correlated with malaria incidence. Between the meteorological variables, RF, T (min.) and RH were negatively associated, while T (max.), T (mean) and WS had positive correlation with the disease incidence.

Table 3 exhibited the statistical association of socio-environmental factors with annual average malaria incidence (2013–2015). The table showed posterior estimates of the results as mean value and standardized division (SD). HDI and PD were inversely while T (min.) directly proportional to malaria incidence during the study period, whereas ES, RF and RH showed no substantive statistical associations with malaria. The DIC value was calculated to be 1344.78.

In Figure 3, the spatial residual variations (structured random effects) were presented as relative risk (RR), taking into account the socio-environmental factors (fixed effect). The map showed that districts with moderate RR (0.75 to 1.24) were present throughout the country with some of the clusters having high RR (1.25 to 1.99) interspersed in between (in KPK. Punjab, Sindh and Balochistan). A few clusters with low RR (0.23 to 0.74) could also be seen in the map.

Besides analyzing the effects of socio-environmental variables on annual average malaria incidence (2013–2015), we also examined the effects of same socio-environmental factors on malaria incidence in individual years during the study period, i.e., 2013, 2014 and 2015. We have attached these results in the Appendix A.

## 4. Discussion

This study explored effects of socio-environmental variables on malaria in Pakistan at the district level. The socio-environmental variables that had a significant effect on malaria incidence in our study included Human Development Index (HDI), minimum temperature (T (min.)) and population density (PD).

Our study confirmed that HDI, and thus socioeconomic status, is inversely proportional to the malaria incidence at the district level, analogous to what predominantly has already been evidenced in the literature [30,31,32]. The low socioeconomic status not only increases the incidence of infectious diseases (e.g., malaria), but also decreases adaptive capacity of individuals against climate changes in the respective geographical areas [33], thereby prolonging their prevalence by increasing vulnerability to disease risks for longer periods [2]. Our study provided evidence that malaria incidence was higher in areas with low socioeconomic status and upholds the notion that malaria is both input and output of poverty; thus, the decision makers in Pakistan should devise strategies for the equitable distribution of the resources and direct malaria control interventions towards high-risk districts having low socio-economic conditions.

Education score (ES) is regarded as specifically an indicator of education status at the district level in Pakistan. Our study, regarded ES as an indirect measurement of the knowledge of parents and children on prevention of common diseases; as, the Pakistan Demographic and Health Survey (PDHS 2012–2013) established that, “knowledge of prevention methods is positively correlated with education and economic status in Pakistan” [34]. The statistical association between ES and malaria incidence was inverse in general, although not statistically significant, during this study; this can be explored in future detailed studies in the individual districts of Pakistan. Earlier studies had suggested that education and socio-economic status were both usually indirectly related with malaria [19,20,35].

The retrospective extrapolations of the population were done at a uniform rate of annual 2.4 percent depreciation at the district level which obviously does not happen in the real world and especially in a thickly inhabited country like Pakistan. In many of the districts in Pakistan, this linear retrospective extrapolation of the population may be an underestimation, while in a few districts may well be an overestimation of the actual population. Pakistan is a densely populated country with average national population density of 166.3/km^2^. In this study, PD exhibited negative relationship with malaria incidence, which is in accordance with the results from most of the former studies. The number of mosquito bites per person are reduced due to the increase in number of individuals residing in same locality, and also due to the fact that urbanization rate has increased over the time leading to reduced number and size of mosquito breeding sites [21,36]. Some other studies found malaria incidence to be positively associated with disease high-risk; in densely populated areas, people tend to have water storage in higher concentrations around them, thereby creating more mosquito breeding sites [37]. This implies that proximity to or remoteness from the waterways (streams, lakes, dams, rivers) may provide variable statistical associations depending upon the study design [36].

Our study could not establish significant statistical relationship of rainfall or relative humidity with malaria incidence. The water reservoirs formed after rainfall, usually, provides suitable environment for mosquito growth, resulting in rise in the malaria incidence; however, the statistical association between rainfall and malaria incidence is not always positive, as excessive rainfall may destroy the mosquito habitats. Rainfall has been found significantly associated with malaria incidence in many of the previous studies, thereby, increasing malaria risk most of the times [11,38,39,40]. RH, generally, has also been directly related with malaria incidence as it facilitates vector longevity [25,41,42]; however, our study could not found any significant statistical association between the two (RH and malaria). T (min.) in our finalized CAR model was found statistically positively associated with the malaria incidence, which is in agreement with the results from most of the preceding studies [25,38,43]. Minimum temperature usually favors malaria, as during the less warm conditions (usually at night) in tropical and temperate areas like Pakistan, people tend to be less covered and mosquito is more active, increasing the chances of bites. Therefore, besides ensuring the access of people to malaria preventive interventions, the relevant authorities also need to work on imparting health education to the commoners in malaria high-risk districts. Such awareness campaigns through different health activities (health education sessions in community meetings and via print and electronic media) can be useful in lowering the malaria risk by enabling people to practice mosquito repellent measures during night time when temperature is relatively low.

The effects of socio-environmental factors on malaria are more evident if we compare malaria incidence map in Figure 2 (without effect of socio-environmental factors) with spatial residual map in Figure 3 (after adding effects of socio-environmental variables). The noticeable is Punjab province that reported the lowest malaria burden in the country, yet in spatial residual maps there could be seen clusters with high RR in some of its southern districts. This may have been due to low socio-economic conditions in these southern districts [44], and/or maybe due to any episode of flash floods or heavy rainfall; however, the temporal and causal relationship cannot be established in our study due to data limitations.

Socio-environmental conditions in Pakistan have affected malaria both directly as well as indirectly by affecting the infrastructure (governance in general and health sector in particular) in the malaria high-risk districts. Socio-environmental factors in our study included socio-demographic and meteorological factors. Infrastructure depends on both of these as it is directly related to socio-economic status and awareness level of a community and can also be affected by the environmental factors such as natural disasters (flood, excessive rainfall or earthquake, etc.) and law and order situations (such as war or unrest). Investing in health and education sectors and developing robust governance system can prove more than beneficial in this regard.

This study has various strengths, as it is the first study to explore the effects of socio-environmental factors on malaria in Pakistan. The Bayesian modeling in exploring malaria dynamics in the country is done for the first time. This study used most reliable data collected through proper channel. The novel information on the socio-environmental drivers of malaria discussed in this study may provide some evidence to the relevant authorities for improvement in current malaria control strategies.

There are several limitations to our study. Aggregated yearly data, rather than monthly or weekly data, were used in this study; thus, the changes of malaria incidence over the time can be masked, and seasonality in malaria incidence could not be established. Detailed information on each case such as age, sex and occupation was not available, and thus not used. The study explained total positive malaria cases and did not discuss species-wise segregation of data. Some of the important socio-environmental variables like vegetation index, mosquito population, elevation and proximity to waterways were not included. This study represented people, which reported to public-sector healthcare facilities only, and gives no information on private-sector malaria burden.

## 5. Conclusions

The study demonstrated that socioeconomic status, human population density and minimum temperature were statistically significantly associated with the annual average (2013–2015) malaria incidence in Pakistan, at the district level. Overall, we asserted in this study that socio-environmental factors play an important role in shaping the malaria dynamics in Pakistan; thus, by including monthly or weekly disease statistics, malaria seasonality across different months within same year needs to be explored in the subsequent studies for malaria hotspots apiece. Discovering local malaria dynamics through the spatial statistical analysis will help policy makers devise evidence-based, most appropriate strategies to reduce malaria risk in the region. Incorporation of geospatial analysis in Pakistan’s malaria control system at all levels is an inevitability. GIS poses unique benefits in the realm of public health; in better monitoring and evaluation, improving disease surveillance, identifying high-risk malaria affected regions, enhancing capacity on adaptation to climate change and extreme weather, and deciding when and where to allocate resources for interventions to realize the goals of malaria control and prevention in the country and the region.

## Figures and Tables

**Figure 1 ijerph-16-01365-f001:**
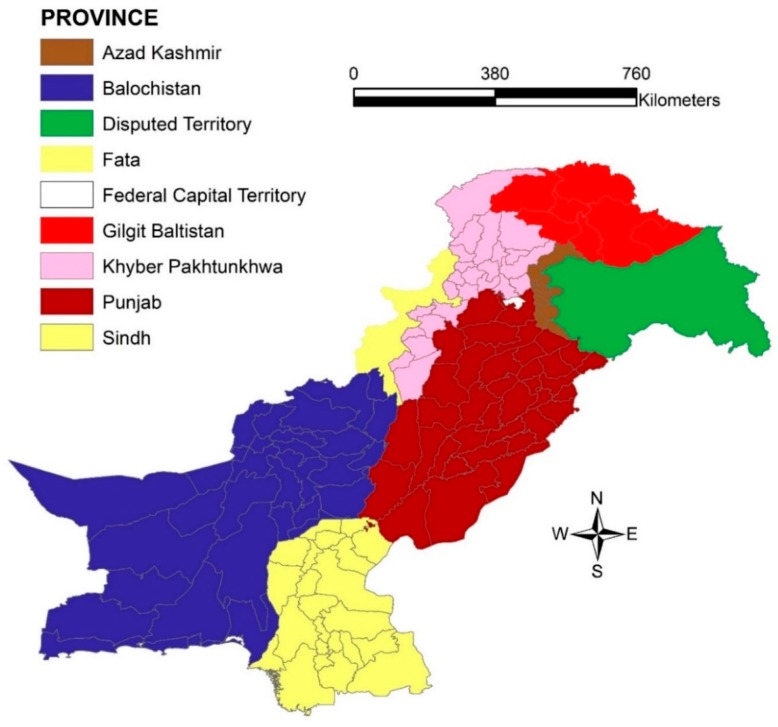
Administrative units (Provinces) in Pakistan.

**Figure 2 ijerph-16-01365-f002:**
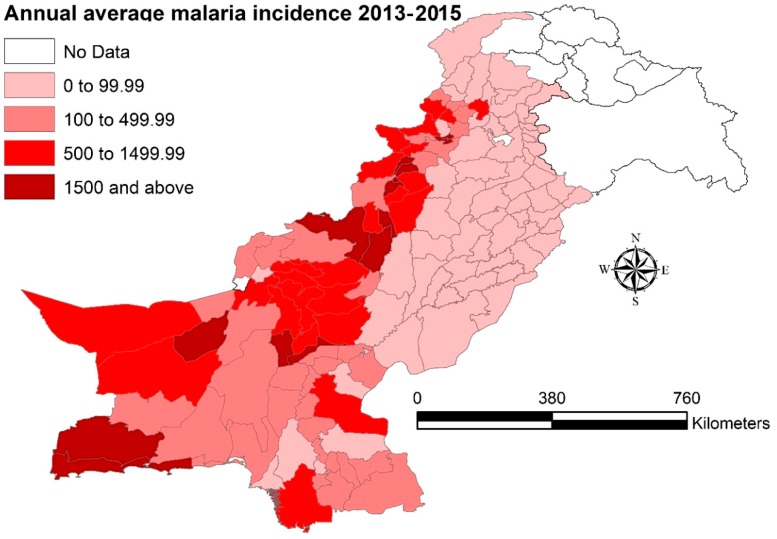
Annual average malaria incidence (per 100,000) at the district level in Pakistan, (2013–2015).

**Figure 3 ijerph-16-01365-f003:**
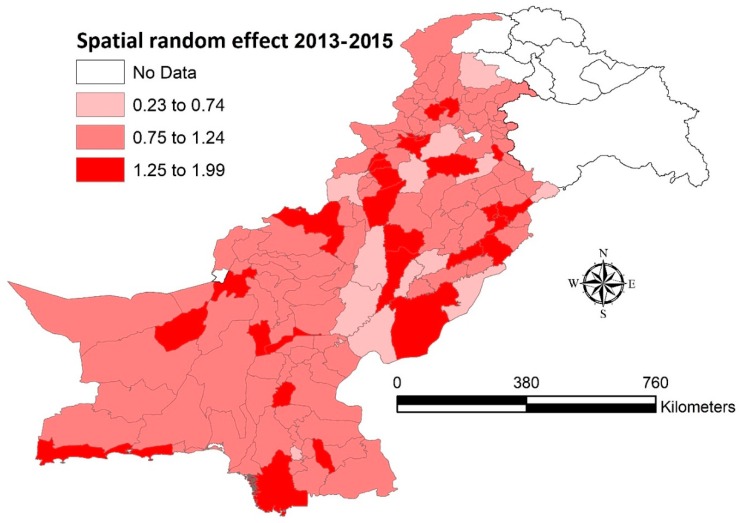
Structured spatial random effects, 2013–2015 (Bayesian CAR model).

**Table 1 ijerph-16-01365-t001:** Descriptive statistics of annual average malaria incidence (per 100,000) and socio-environmental variables at the district level in Pakistan (2013–2015).

Variables	Mean	Standard Deviation	Minimum	Quantiles			Maximum
				25	50	75	
Incidence	517.93	1239.88	0.04	6.02	119.23	564	11,565.23
HDI	0.46	0.13	0.13	0.38	0.48	0.56	0.71
ES	60.89	14.09	29.44	49.99	62.17	72.82	84.85
PD	313.54	438.27	4	57	224.5	444.5	3566
RF	515.18	346.5	118.16	216.25	418.33	744.23	1462.54
T (min.)	16.18	3.21	8.2	14.04	16.17	18.45	22.91
T (max.)	29.52	2.62	23.42	27.95	29.58	31.35	34.14
T (mean)	22.85	2.87	15.81	21.18	22.83	24.82	28.52
RH	55.87	5.66	41.88	51.36	57.28	60.84	63.67
WS	3.61	1.59	2.13	2.42	2.87	4.59	8.34

Note: Each variable in the table represents annual average for 136/146 districts in Pakistan. HDI (human development index), ES (education score), PD (population density, per km^2^), RF (rainfall, mm), T (min.) (minimum temperature, °C), T (max.) (maximum temperature, °C), T (mean) (mean temperature, °C), RH (relative humidity, %), WS (wind speed km/h).

**Table 2 ijerph-16-01365-t002:** Spearman’s correlation between annual average malaria incidence and socio-environmental variables at the district level in Pakistan (2013–2015).

Variables	Incidence	HDI	ES	PD	RF	T (min.)	T (max.)	T (mean)	RH	WS
HDI	−0.730 **	1								
ES	−0.652 **	0.678 **	1							
PD	−0.618 **	0.518 **	0.685 **	1						
RF	−0.302 **	0.257 **	0.510 **	0.373 **	1					
T (min.)	−0.004	−0.014	−0.087	0.013	−0.772 **	1				
T (max.)	0.217 *	−0.201 *	−0.367 **	−0.246 **	−0.926 **	0.916 **	1			
T (mean)	0.092	−0.098	−0.217 *	−0.104	−0.854 **	0.983 **	0.971 **	1		
RH	−0.366 **	0.284 **	0.647 **	0.529 **	0.804 **	−0.297 **	−0.589 **	−0.431 **	1	
WS	0.179 *	−0.088	−0.432 **	−0.387 **	−0.923 **	0.637 **	0.805 **	0.723 **	−0.841 **	1

Note: ** Correlation is significant at the 0.01 level (two-tailed), * Correlation is significant at the 0.05 level (two-tailed). Each variable in the table represents annual average for 136/146 districts in Pakistan. HDI (human development index), ES (education score), PD (population density, per km^2^), RF (rainfall, mm), T (min.) (minimum temperature, °C), T (max.) (maximum temperature, °C), T (mean) (mean temperature, °C), RH (relative humidity, %), WS (wind speed km/h).

**Table 3 ijerph-16-01365-t003:** Statistical association between socio-environmental factors and annual average malaria incidence at the district level in Pakistan (Bayesian conditional autoregressive (CAR) model), 2013–2015.

Variables	Mean	SD	MC error	2.50%	Median	97.50%
HDI	−3.965	1.127	0.05994	−6.297	−3.905	−1.754
ES	−0.02151	0.01164	6.23 × 10^−4^	−0.04017	−0.02443	2.44 × 10^−4^
PD	−7.41 × 10^−4^	3.25 × 10^−4^	1.59 × 10^−5^	−0.001406	−7.41 × 10^−4^	−1.05 × 10^−4^
RF	0.001226	0.001124	6.00 × 10^−5^	−0.001657	0.001446	0.002826
T (min.)	0.1398	0.04885	0.002618	0.05275	0.1501	0.2145
RH	0.003186	0.01803	9.67 × 10^−4^	−0.02377	−1.80 × 10^−4^	0.03471

Note: Each variable in the table represents annual average for 136/146 districts in Pakistan. MC (Monte Carlo), SD (standardized division), HDI (human development index), ES (education score), PD (population density, per km^2^), RF (rainfall, mm), T (min.) (minimum temperature, °C), RH (relative humidity, %).

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
