# Peer review of "Effects of Socio-Environmental Factors on Malaria Infection in Pakistan: A Bayesian Spatial Analysis"

_ijerph, 2019, doi:10.3390/ijerph16081365_

Round 1

Reviewer 1 Report

The authors present an analysis of malaria human cases in Pakistan and they identify statistical associations with some socieconnomic and environmental factors using a spatial Bayesian conditional autoregressive model. The approach is one of the classic ones used in this kind of questions and belongs to the robust ones at least with respect to the type I error. The manuscript is well written and the reader can follow without difficulties.

Major comments

The authors refer throughout the text to associations although they find statistical associations. This creates the impression that they identify causality which is not the case. This should be corrected.

Some comments:

Abstract.

 l. 24-26 The numbers presented are not straitforward undertstandable. What does teh star mean after the -7.41? The numbers after the period deffer and create the impression of a precision that is not given.

Please clarfiy that you find statistical associations and not causal associations.Please mantion the major limitation of the study in the abstract.

Introduction.  No comment

Methods.

l. 74 Can you please explain why do you use the data of the years 2013-2015 and not a longer period, e.g. 2011-2015.

l. 82-83 The authors should discuss the retrospective extrapolation they perform to estimate the population size of 2013 2014 2015. In countries like Pakistan population increases fast and not necessarily linear. The average may be underestimation. This point should be mentioned in the Discussion section of the manuscript.

l. 120 The authors should explain better why they chose to check multicollinearity by a simple Spearman correlation coefficient and why they chose the threshold of 0.8 which is still high. There are other more robust indicators such as the variation inflation factor. Collinearity is crucial in their approach and should be checled carefully. Please elaborate on that.

l. 174 ‘standardised Division’ I thin khte authors mean Standard Deviation.

L. 231 is typical for the rest of the text. The authors talk about association bu they find or don’t find statistical associations. This should be corrected in the whole manuscript.

l. 260 ‘ The study established…’. This is a strong statement. The authors simply find a few indicative associations;  useful but this is far from being an certainty.

l. 358 Reference 12 is unclear. Please correct

Author Response

Comments and Suggestions for Authors

The authors present an analysis of malaria human cases in Pakistan and they identify statistical associations with some socieconnomic and environmental factors using a spatial Bayesian conditional autoregressive model. The approach is one of the classic ones used in this kind of questions and belongs to the robust ones at least with respect to the type I error. The manuscript is well written and the reader can follow without difficulties.

Reply: We thank reviewer for these encouraging words.

Major comments

Q 1. The authors refer throughout the text to associations although they find statistical associations. This creates the impression that they identify causality which is not the case. This should be corrected.

 Reply: Yes, the reviewer has rightly identified the mistake and we have rectified this throughout the document by adding word “statistically” wherever needed; e.g., we wrote,

“statistically positively associated”, (Lines 255 to 256 in the revised document)

And

 “statistically significant” (Line 228 in the revised document)

And

“statistically significantly associated” (Line 297 in the revised document)

Some comments:

Abstract.

Q 2. l. 24-26 The numbers presented are not straitforward undertstandable. What does teh star mean after the -7.41? The numbers after the period deffer and create the impression of a precision that is not given.

Reply: a) Although we tried to be as clear as possible, yet the reviewer’s concern is not far from truth. The limitation for word count has let us write as precise yet clear as possible. In case of any ambiguities the details can be checked in the Table 3 of the revised document.

b) The asterisk sign “ * ” has been replaced by multiplication sign “ X ”  (Line 27 of the revised document).

Q 3. Please clarfiy that you find statistical associations and not causal associations. Please mantion the major limitation of the study in the abstract.

Reply: a) Yes, we found only statistical associations and not the causal associations and we have rectified this throughout the document as explained in the reply to Q 1 major comment asked by the reviewer.

b) Yes, we have mentioned the major limitation in the abstract and following sentence was added,

A major limitation was the use of yearly, rather than monthly/weekly malaria data in this study”. (Line 18 to 19 in the revised document)

Moreover, to meet the requirements of maximum 200 word count in the abstract, Line 18 to 20 in the revised document were rephrased and adjusted.

Introduction.  No comment

Methods.

Q 4. l. 74 Can you please explain why do you use the data of the years 2013-2015 and not a longer period, e.g. 2011-2015.

Reply: We have addressed this question and added following sentences,

The vector-borne diseases exhibit periodicity due to endogenous and exogenous factors, and literature shows that periodicity due to extrinsic factors (such as socio-climatic factors) is shorter than that due to the intrinsic ones (factors intrinsic to the parasites) [27], thus, we applied Bayesian statistical analysis on the three years’ combined annual average data so that this period was neither too short that it may mimic the results from any single year malaria data analysis (2013, 2014 or 2015), nor was too long that the periodicity due to extrinsic factors (socio-climatic factors) may be masked by the periodicity due to intrinsic factors”. (Line 111 to 117 in the revised document)

Q 5. l. 82-83 The authors should discuss the retrospective extrapolation they perform to estimate the population size of 2013 2014 2015. In countries like Pakistan population increases fast and not necessarily linear. The average may be underestimation. This point should be mentioned in the Discussion section of the manuscript.

Reply: Yes, the reviewer has mentioned an important point and we have added following sentences in the discussion part,

“The retrospective extrapolations of the population were done at a uniform rate of annual 2.4 percent depreciation at the district level which obviously does not happen in the real world and especially in a densely inhabited country like Pakistan. In many of the districts in Pakistan, this linear retrospective extrapolation of the population may be underestimation, while in a few districts may well be overestimation of the actual population”. (Line 232 to 236 in the revised document)

Q 6. l. 120 The authors should explain better why they chose to check multicollinearity by a simple Spearman correlation coefficient and why they chose the threshold of 0.8 which is still high. There are other more robust indicators such as the variation inflation factor. Collinearity is crucial in their approach and should be checked carefully. Please elaborate on that.

Reply: We chose to check multicollinearity by Spearman’s correlation coefficient based on literature review as there were many studies done in the past that followed the same criteria (e.g., Zhang et al. IJERPH, 2018, and Qi et al. BMC Public Health 2014). The same threshold was followed by the above mentioned studies, however, our criteria was not merely based on multicollinearity alone but we checked for lower DIC values for the different combinations of independent variables to choose our final model. The reviewer has suggested other indicators like variance inflation factor, which we shall consider using in the future research.

Q 7. l. 174 ‘standardised Division’ I think the authors mean Standard Deviation.

Reply: This is actually “Standardized Division” and not the “Standard Deviation”. In our study, the posterior estimates of the results were shown as mean value and standardized division (SD).

Q 8. L. 231 is typical for the rest of the text. The authors talk about association but they find or don’t find statistical associations. This should be corrected in the whole manuscript.

Reply: Yes, we have corrected this throughout the revised document

Q 9. l. 260 ‘ The study established…’. This is a strong statement. The authors simply find a few indicative associations;  useful but this is far from being an certainty.

Reply: Yes, we agree to the reviewer and we have replaced the word “established” with “demonstrated” (Line 296 in the revised document)

Q 10. l. 358 Reference 12 is unclear. Please correct

Reply: Thank you for pointing out, we have corrected this reference in the revised document. (Line 394 to 395 in the revised document)

Reviewer 2 Report

What is the definition and contents of socio-environment factors in your study?  

What are the factors to influence the incidence of malaria in Pakistan? 

What is the relationship between the socio-environmental factors and occurrence of malaria? How do you use your findings to prevent of control the occurence of malaria?  

What is the implication of HDI and district education score ?

What is the relationship between infrstructure and socienviromental factors in this study?     

Author Response

Comments and Suggestions for Authors

Q 1. What is the definition and contents of socio-environment factors in your study?  

Reply: As it has been demonstrated in literature since long that socio-environmental variables play complex and inconsistent role in shaping up malaria dynamics in different geographic settings and that its contents are also varying according to different geographical areas so because of its subjectivity we chose not to define the term, “socio-environmental factors” in our study; however we already explained its contents in the context of our study which we mentioned in the sentences as follows,

“…. socio-environmental (socio-demographic such as human development index, education ranking, and population density) and (meteorological variables such as temperature, rainfall, and relative humidity)….” (Line 76 to 78 of the revised document)

Q 2. What are the factors to influence the incidence of malaria in Pakistan? 

Reply: We have added following sentences in the revised document,

“Erratic malaria transmission patterns due to various factors like climatic changes, natural calamities (frequent flooding and irregular rainfall pattern), resource constraints, weak health infrastructure, poor socio-economic conditions, domestic unrest in many of the districts in FATA and Balochistan, limited access to healthcare delivery services and frequent migrations of people within the country and in the Pak-Afghanistan bordering area are amongst the key reasons for unstable malaria incidence in Pakistan” (Line 61 to 66 in the revised document).

Q 3. What is the relationship between the socio-environmental factors and occurrence of malaria? How do you use your findings to prevent of control the occurence of malaria?  

Reply: a) The relationship between socio-environmental factors and occurrence of malaria has already been mentioned in the manuscript

Socio-environmental variables influence spatial and temporal distribution, intensity and duration of infectious diseases in general [4-6]; and occurrence, transmission, seasonality and periodicity of malaria in particular [7-9]. Malaria dynamics are better understood in the perspective of native socioeconomic and climatic conditions to discover the role of local factors influencing malaria variability in space and time [10].

Meteorological factors, e.g., rainfall, temperature and humidity, are established to having associations with malaria incidence from temporal and spatial perspectives [9,11-14]. These meteorological factors when co-act synergistically, increase the duration of larvae development, shorten the incubation period of parasites, prolong mosquito survival, provide a favorable swampy habitat to the vectors, and increase number of mosquitoes and their bites, thus positively related to malaria high-risk [15-18].

Malaria and poverty have long been linked together; socioeconomic status along with education are believed to be indirectly related to malaria, and improvement in both these factors has contributed in disease control in many developing countries in recent times [19,20]. Human population density, like other socio-environmental factors contribute to heterogeneity in malaria infection, is indirectly related to entomological inoculation rate (EIR) thereby reducing the risk of malaria in areas with high human population densities [21]”. (Line 41 to 57 in the revised document)

b) The findings in our study suggested that malaria is statistically associated with socio-economic conditions and with minimum temperature at the district level in Pakistan.

Keeping in view the findings in this study, we have added following sentences in the revised manuscript

“so, the decision makers in Pakistan should devise strategies for the equitable distribution of the resources and direct malaria control interventions towards high-risk districts having low socio-economic conditions”. (Lines 220 to 222 in the revised document)

And,

“Therefore, besides ensuring the access of people to malaria preventive interventions, the relevant authorities also need to work on imparting health education to the commoners in malaria high-risk districts. Such awareness campaigns through different health activities (health education sessions in community meetings and via print & electronic media) can be useful in lowering the malaria risk by enabling people to practice mosquito repellent measures during night time when temperature is relatively low” (Lines 259 to 264 in the revised document).

Besides, we have also mentioned the importance of further detailed public health research in the individual hotspots apiece to explore more about malaria dynamics in the country, as follows;

“Overall, we asserted in this study that socio-environmental factors play an important role in shaping the malaria dynamics in Pakistan, thus, by including monthly or weekly disease statistics, malaria seasonality across different months within same year needs to be explored in the subsequent studies for malaria hotspots apiece. Discovering local malaria dynamics through the spatial statistical analysis will help policy makers devise evidence-based, most appropriate strategies to reduce malaria risk in the region” (Line 298 to 303 in the revised document).

Q 4. What is the implication of HDI and district education score ?

Reply: HDI and district education score were used for socio-economic status and indirect measurement of knowledge on preventive methods respectively, and we have already discussed the implication of HDI and district education score in the manuscript, e.g.,

HDI in our study was used to describe the socio-economic status for respective districts in the country. HDI is summary measure of human development in three basic dimensions, i.e health (life expectancy at birth), education (years of education received by people aged 25 years and older) and standard of living (measured by Gross National Income, GNI per capita). Experts consider HDI as arguably a better development indicator than Gross Domestic Product (GDP) as it explains how rich the lives of people are, rather than how rich the economy is. District wise HDI were downloaded from United Nations Development Program Pakistan website (www.pk.undp.org). ES is the district-wise education score in Pakistan (arithmetic average of, i) enrolment, ii) learning outcomes, iii) retention and iv) gender parity scores; giving equal weight to each of these indicators) and is downloaded from “Alif Ailaan” website (www.alifailaan.pk/district_rankings), which is a non-profit organization working in the field of education in Pakistan. (Line 93 to 103 in the revised document)

And

“Our study confirmed that HDI and thus socioeconomic status is inversely proportional to the malaria incidence at the district level, analogous to what predominantly has already been evidenced in the literature [29-31]. The low socioeconomic status not only increases the incidence of infectious diseases (e.g., malaria) but also decreases adaptive capacity of individuals against climate changes in the respective geographical areas [32], thereby, prolonging their prevalence by increasing vulnerability to disease risks for longer periods [31]. Our study provided evidence that malaria incidence was higher in areas with low socioeconomic status and upheld the notion that malaria is both input and output of the poverty at the same time so, the decision makers in Pakistan should devise strategies for the equitable distribution of the resources and direct malaria control interventions towards high-risk districts having low socio-economic conditions.

Education score (ES) is regarded as specifically an indicator of education status at the district level in Pakistan. Our study, regarded ES as an indirect measurement of the knowledge of parents and children on prevention of common diseases; as, the Pakistan Demographic and Health Survey (PDHS 2012-13) established, that, “knowledge of prevention methods is positively correlated with education and economic status in Pakistan” [33]. The statistical association between ES and malaria incidence was inverse in general, although not statistically significant, during this study; and this can be explored in future detailed studies in the individual districts of Pakistan. Earlier studies had suggested that education and socio-economic status were both usually indirectly related with malaria.” (Lines 213 to 231 in the revised document)

And

“…the relevant authorities also need to work on imparting health education to the commoners in malaria high-risk districts. Such awareness campaigns through different health activities (health education sessions in community meetings and via print & electronic media) can be useful in lowering the malaria risk…..” (Line 259 to 261 of revised document)

Q 5. What is the relationship between infrstructure and socienviromental factors in this study?     

Reply: The relationship between infrastructure and socio-environmental factors has been added in the revised document in Lines…to…,

 “Socio-environmental conditions in Pakistan have affected malaria both directly as well as indirectly by affecting the infrastructure (governance in general and health sector in particular) in the malaria high-risk districts. Socio-environmental factors in our study included socio-demographic and meteorological factors. Infrastructure depends on both of these as it is directly related to socio-economic status and awareness level of a community and can also be affected by the environmental factors like natural calamities (flood, excessive rainfall or earthquake etc) and law and order situations (such as war or unrest). Investing in health and education sectors and developing robust governance system can prove more than beneficial in this regard”. (Line 273 to 280 in the revised document)

Round 2

Reviewer 2 Report

The manuscript has revised as comments of reviewer. It got much improvement compared to the original version.